

# Investigation and research on elderly people's willingness to combine medical and health care and related factors in coastal cities in eastern China

Yehong Wei[1], Yanxiang Sun[2], Yuting Li[2], Xufeng Chen[3], Yingyu Wu[2], Xindi Fang[2] and Ruichen Zhu[4,5]

[1] The Second Affiliated Hospital of Zhejiang Chinese Medical University, Hangzhou, China
[2] Zhejiang Chinese Medical University, Hangzhou, China
[3] Hangzhou Hospital of Traditional Chinese Medicine, Hangzhou, China
[4] Zhejiang Hospital, Hangzhou, China
[5] Affiliated Zhejiang Hospital Zhejiang University School of Medicine, Hangzhou, China

Corresponding author
Ruichen Zhu, 490832240@qq.com

## ABSTRACT

**Background:** The problem of global aging was becoming increasingly prominent. At present, the empty nest and miniaturization of family structure reduce the function of home-based elderly care.

**Methods:** A questionnaire survey was conducted on 347 elderly people in multiple communities and nursing homes in eastern coastal cities of China, and 13 institutional staff members of eight nursing institutions that carried out the medical-nursing integration model were interviewed as the research objects. The survey mainly focuses on the basic characteristics of the elderly, the family support system, and the acceptance of medical care and health care. The influencing factors were screened by t test, univariate analysis and multivariate logistic regression analysis. SPSS software was used to test the reliability and validity of the questionnaire, and the Crobach's was 0.792, which can be considered that the questionnaire had good internal reliability. The classification of the questionnaire was reasonable, the reliability of the questionnaire was high, and the internal consistency of the scale was high. According to KMO and Bartlett test, KMO = 0.826, $\chi 2$ = 853.731, the degree of freedom was 36, and the $P$ value was 0.000.

**Results:** The proportion of male and female respondents was 48.1% and 51.9% respectively. Multivariate logistic regression analysis results showed that gender had no statistical significances on the degree of support for combine medical and health care in the elderly ($P > 0.05$). The results showed that gender, age, marital status, medical insurance type and old-age insurance type had little effect on the support of the combination of medical care and health care for the elderly ($P > 0.05$). Compared with the control group with education below primary school, the elderly with bachelor's degree or above are more willing to support the combination of medical care and health ($P < 0.05$). The registered residence type is more obvious than that of the urban residents ($P < 0.05$). Compared with the enterprise employees in the control group, the elderly who were employed as migrant workers before retirement were more willing to support the combination of medical care and health ($P < 0.05$). From the perspective of family monthly income, the elderly with family income ≥10,000 RMB have more obvious support for the combination of medical care and

health than the elderly with family monthly income <3,000 RMB in the control group ($P < 0.05$). In terms of the degree of understanding, the degree of understanding and support in different degrees are significantly higher than that in the control group ($P < 0.05$).

**Conclusion:** Through multivariate logistic regression analysis, education level, registered residence, pre-retirement occupation and family income are more obvious for the elderly to support medical care and health. It is necessary to increase investment in elderly activity centers, actively carry out activities.

## INTRODUCTION

At present, the problem of global aging was becoming more and more prominent, and our country was also in a period of rapid development of population aging. Up to November 2020, there were approximately 264 million people over 60 years old in my country, accounting for 18.7% of the total population. It is estimated that by 2025, the number of elderly people aged 65 and above in China will reach 200 million, and the aging of China's population is deepening, and the trend will be irreversible (*Han, 2022*; *None, 2021*).

At present, indicators such as average life expectancy and mortality are still an important basis for measuring health status in China (*Kang et al., 2021*; *Si et al., 2021*). With the aging and disability of the population becoming more and more prominent, it is still extremely challenging to meet the needs of elderly care services and medical care, increase the life expectancy of the population, reduce the mortality rate, and promote national health. The combination of medical and elderly care in various countries has carried out innovations to varying degrees according to the actual situation. Most of them are integrated care that combines comprehensive medical care, mental health and social services (*Arku et al., 2022*). Such as Program of All-inclusive Care for the Elderly (PACE), ASAP in the United States, EHCH in England, *etc.* In China, the traditional community-based home-based elderly care and most elderly care institutions are mainly based on life care, lack of one-stop medical care and health care services, and cannot fully meet the medical service needs of the elderly. Under this premise, the elderly have increased demand for medical care and health care, and the governments of coastal cities in eastern China are also demand-oriented, and actively promote the construction of integrated medical care and health care projects such as home care, community care, and day care. "Medical" includes outpatient services and rehabilitation physiotherapy, "Health care" includes daily care and emotional support (*Zhang et al., 2017*). Combine medical and health care advocates the early intervention and integration of medical resources into elderly care services, so as to meet the needs of elderly care services and medical services for the elderly, while promoting national health. However, the implementation of the integration project of medical care and health is affected by the government level and the personal will of the elderly (*Wang et al., 2021*, *2022*). At the government level, the smooth

connection between medical and health care and elderly care services is insufficient, the quality of medical care and elderly care services needs to be improved, and relevant economic and technical policy support needs to be improved, which affect the construction of combine medical and health care projects. In terms of personal wishes, the elderly are more likely to choose home care. However, the empty nest and miniaturization of the current family structure reduces the function of home care; at the same time, the increase in the family structure of "4-2-1", "4-2-2" and even "4-2-3" caused by the birth policy has caused certain obstacles to the advancement of combine medical and health care (*Liu et al., 2019*). Therefore, we aimed to study elderly people's willingness to combine medical and health care and related factors in coastal cities in eastern China and to analyze the relevant factors, in order to build a local integrated service innovation model of combine medical and health care, which are suitable for the social conditions and strategic and institutional background of coastal cities in eastern China.

# OBJECTS AND METHODS

## Research object
This research was based on two typical communities and eight nursing homes in eastern coastal cities in China, one of the first batch of 50 "national pilot units for the combination of medical care and maintenance" were used as research carriers. A total of 354 questionnaires were distributed, and 347 were actually recovered, with a recovery rate of 98.0%. Inclusion criteria: (1) age ≥ 60 years; (2) Aware and able to complete the questionnaire independently or with the assistance of others; (3) Residence in eastern coastal cities in China ≥ 1 year.

## Ethical statement
This study was approved by the Ethics Committee of Zhejiang Hospital (2020, No. 115 and 2021, No. 75k), and all elderly people had signed informed consent forms.

## Research methods
In June 2021, through a large number of literature review, a questionnaire was formulated for the medical care of the willingness and influencing factors of the elderly in Eastern coastal cities in China. With the strong support of the school of Humanities and management of Zhejiang Chinese Medical University a college student practice group was established to investigate this theme. Before this investigation, the members of the research group shall conduct unified training for the members of the practice group to ensure that the members of the practice group use unified guidelines. The survey was conducted in some communities and nursing homes in Eastern coastal cities in China by convenient sampling. The survey was conducted by questionnaire and written questionnaire. For the elderly who can independently operate the mobile phone questionnaire star, the questionnaire star method was adopted for the survey. For the elderly who cannot use the mobile phone to answer, the members of the practice group conducted the survey in the form of question and answer.

## Research tools

This study followed the relevant spirit of the State Council's "guiding opinions on promoting the combination of medical and health care and elderly care services" (*Han & Li, 2018*) and "Hangzhou measures for promoting the integration of medical and health care and intelligent medical services" (*None, 2015*). Referring to the ADL self-care theory, the members of the research group compiled a questionnaire on the willingness of the elderly in Eastern coastal cities in China to combine medical care and health and related factors. The questionnaire survey included general information, mental health, physical status, family support system, acceptance of the integration of medical care and health, *etc.* The preparation process of the questionnaire was as follows: Through a large number of literature review, the interview outline of the influencing factors of medical maintenance and health was formulated. Eight elderly people were interviewed in the Second Affiliated Hospital of Zhejiang Chinese Medical University and Zhejiang hospital to collect the core information of the elderly about medical care and health. Through SPSS software to test the reliability and validity of the questionnaire, it is concluded that Crobach's was 0.792, which can be considered that the questionnaire had good internal reliability.

The classification of the questionnaire was reasonable, the reliability of the questionnaire was high, and the internal consistency of the scale was high. According to KMO and Bartlett test, KMO = 0.826, $\chi^2$ = 853.731, the degree of freedom was 36, and the *P* value was 0.000. It showed that the correlation between various factors was very strong, and the results were scientific and reliable.

## Quality control

In July 2021, with the support of the School of Humanities and Management of Zhejiang University of Traditional Chinese Medicine, a college student practice group named "Beautiful Zhejiang Practice Group" was established to conduct research on this theme. Before the official start of the research, the members of the practice group conduct unified training to ensure that the members of the practice group use unified guidelines to explain the research purpose and filling method to the research subjects when collecting data, so as to gain the trust and cooperation of the research subjects. During the investigation, all questionnaires and interviews were conducted independently and anonymously, without involving the personal privacy of the respondents and the interviewees, and the true thoughts of the respondents were collected as much as possible. During the data entry process, data checkers repeatedly analyze and proofread the data to ensure the accuracy and reliability of the data.

## Pilot scale study

Before the formal investigation, a representative community in the coastal cities and municipalities in eastern China that has developed a model of combining medical care, nursing and health care—Fuyang Ruifeng Senior Apartment Community, and two elderly hospitals—Zhejiang Hospital and Xihu District Integrated Traditional Chinese and Western Medicine Hospital were selected. A 3-day pre-survey was conducted among 90 senior citizens. At the same time, 13 institutional staff members from four elderly care

institutions of different levels and types were selected for interviews, considering the heterogeneity factors such as geographical location and scale. According to the results of the pre-investigation and the interview content of the staff, the content of the questionnaire was modified and deleted, and the structure was adjusted so that the elderly could better understand the questions and answer them independently.

## Sample size calculation

This survey uses convenience sampling to select two typical communities and eight nursing homes in eastern coastal cities and cities in China, one of the first batch of 50 "national pilot units for the combination of medical care and maintenance", as research carriers. In order to obtain higher accuracy, the confidence level was determined as 95% (=1.962), the maximum allowable absolute error was 8% ($\mu = 0.08$). According to the maximum value of $P = 0.5$, the initial sample size was determined as 286, and 354 questionnaires were actually recovered. A total of 347 copies, with a recovery rate of 98.0%. To ensure the accuracy of the data, the investigators uploaded the survey data in real time on the terminal.

## Statistical analysis

The data set was established with Excel 2017 and imported into SPSS 21.0 software package for statistical analysis. The frequency, composition ratio and are used for statistical description. T-test, one-way ANOVA and multiple linear regression analysis were used to analyze the influencing factors of family medical maintenance integrated service demand. The difference was statistically significant ($P < 0.05$).

# RESULTS

## Basic patient information

According to the purpose and object of the survey, the basic information of the surveyed elderly, such as gender, age, education level, marital status, medical insurance, pension insurance type and monthly family income, were analyzed as follows (Fig. 1). A total of 48.1% of men and 51.9% of women participated in the questionnaire. Among them, 37.5% were aged 60–65, 13.3% were aged 66–70, 15.6% were aged 71–75, 9.5% were aged 76–80 and 24.2% were aged over 80. The number of people with education level of primary school and below accounted for 27.7%, that of junior high school and senior high school accounted for 41.8%, and that of college and above accounted for 30.6%. 76.9% were married, 2% were unmarried, 17.6% were widowed and 3.5% were divorced. The number of people without medical insurance accounted for 6.6%, and the number of people with commercial medical insurance accounted for 9.8%. The new rural cooperative insurance accounted for 21%, the medical insurance for urban residents accounted for 14.1% and the medical insurance for urban employees accounted for 42.7%. Both commercial insurance and medical insurance accounted for 5.8%. The number of people without endowment insurance accounted for 15.3%, the number of people whose endowment insurance was government organs and institutions accounted for 33.7%, urban employees or urban and rural residents account for 36%, land expropriated farmers accounted for 7.5%, and

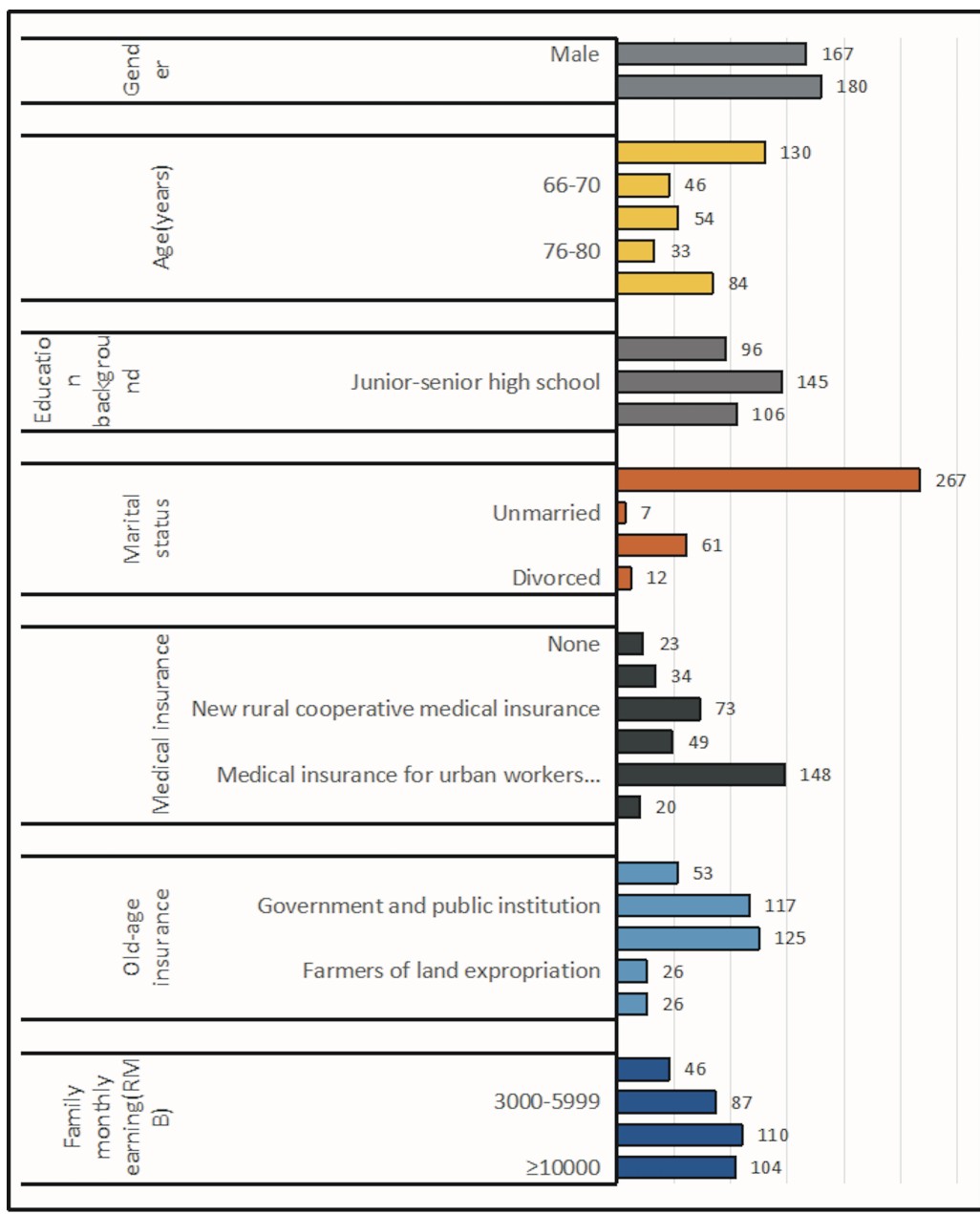

**Figure 1  Basic information of the elderly.**

commercial insurance accounted for 7.5%. 13.3% of households have a monthly income of less than 3,000 RMB, 25.1% have a monthly income of 3,000–5,999 RMB, 31.7% have a monthly income of 6,000–9,999 RMB, and 30% have a monthly income of 10,000 RMB or more.

## Residents' cognition of the combination of medical and health care

Residents have a high level of awareness of the combination of medical care and health. Well known, know and better known (63.4%), little known and unknown (35.6%).

The cognitive level of residents aged 60–80 was more than 60%, and the cognitive level of residents over 80 was lower than that of other age groups ($\chi2 = 43.925$, $P < 0.05$). The higher the education level of the elderly, the higher the recognition of the medical and health care ($\chi2 = 65.490$, $P < 0.05$). Registered residence type, urban residents' awareness of medical and health care combination was higher than that of suburban residents ($\chi2 = 28.331$, $P < 0.05$). The overall cognitive level of married residents was significantly higher than that of residents with other marital status ($\chi2 = 32.027$, $P < 0.05$). Residents whose occupation before retirement was enterprise employees and national civil servants have higher cognition ($\chi2 = 51.861$, $P < 0.05$). Residents whose medical insurance was commercial insurance or commercial insurance and medical insurance have higher overall cognition than residents with other medical insurance ($\chi2 = 64.252$, $P < 0.05$). Residents without pension insurance have higher awareness than other residents ($\chi2 = 27.857$, $P < 0.05$). The higher the monthly family income, the higher the residents' cognition ($\chi2 = 43.312$, $P < 0.05$) (Table 1).

## Family situation

Residents' overall support for the combination of medical and health care was relatively high. The family monthly income was higher, the residents' support for the combination of medical and health care was higher. The residents willing to spend 1,000–1,999 RMB and ≥4,000 RMB per month for the combination of medical and health care have a higher degree of support than other spending levels. In the survey, seven elderly people have no children. After excluding seven the elderly, the data showed that 98.5% of their children support, support and very much support the combination of medical care and health. The higher level of support of children, the higher level of support of the elderly (Table 2).

## Willingness to combine medical care with health care

More than 95% of the elderly and their children supported the combination of medical and health care. In the survey, the higher the understanding of the combination of medical and health care, the higher the degree of support ($\chi2 = 49.694$, $P < 0.05$) (Table 3, Fig. 2).

## Multivariate logistic regression analysis

Based on the $\chi2$ test, this study combined with multivariate ordered logistic regression analysis. To explore the influence of various factors on the willingness of the elderly to combine medical and health care in Eastern coastal cities in China, a multi-factor ordered logistic regression analysis was carried out on 347 cases. The results showed that gender, age, marital status, medical insurance type and old-age insurance type had little effect on the support of the combination of medical and health care for the elderly ($P > 0.05$). In terms of educational level, compared with the control group with education below primary school, the elderly with bachelor's degree or above are more willing to support the combination of medical and health care ($P < 0.05$). The registered residence type was more obvious than that of the urban residents ($P < 0.05$). In terms of occupation, compared with the enterprise employees in the control group, the elderly who were employed as migrant workers before retirement were more willing to support the combination of medical and

**Table 1 The relationship between personal general condition and cognitive condition.**

| Items | Number | Cognition degree | | | | | χ² value | P value |
|---|---|---|---|---|---|---|---|---|
| | | Well known | Know | Better known | Little known | Unknown | | |
| Sex | | | | | | | 8.836 | 0.087 |
| Male | 167 | 16 (9.6%) | 50 (29.9%) | 51 (30.5%) | 28 (16.8%) | 22 (13.2%) | | |
| Female | 180 | 20 (11.1%) | 46 (25.6%) | 37 (20.6%) | 37(20.6%) | 40 (22.2%) | | |
| Age (years) | | | | | | | 43.923 | 0.000 |
| 60–65 | 130 | 17 (13.1%) | 39 (30.0%) | 38 (29.2%) | 25 (19.2%) | 11(8.5%) | | |
| 66–70 | 46 | 6 (13.0%) | 14 (30.4%) | 16 (34.8%) | 5 (10.9%) | 5 (10.9%) | | |
| 71–75 | 54 | 3 (5.6%) | 14 (25.9%) | 18 (33.3%) | 12 (22.2%) | 7 (13.0%) | | |
| 76–80 | 33 | 5 (15.2%) | 9 (27.3%) | 5 (15.2%) | 6 (18.2%) | 8 (24.2%) | | |
| >80 | 84 | 5 (6.0%) | 20 (23.8%) | 11 (13.1%) | 17 (20.2%) | 31 (36.9%) | | |
| Level of education | | | | | | | 65.490 | 0.000 |
| Primary school the following | 44 | 2 (4.5%) | 7 (15.9%) | 6 (13.6%) | 9 (20.5%) | 20 (45.5%) | | |
| Primary school | 52 | 1 (1.9%) | 9 (17.3%) | 17 (32.7%) | 9 (17.3%) | 16 (30.8%) | | |
| Junior high school | 69 | 9 (13.0%) | 18 (26.1%) | 12 (17.4%) | 15 (21.7%) | 15 (21.7%) | | |
| Technical secondary school | 16 | 0 (0.0%) | 6 (37.5%) | 4 (25.0%) | 5 (31.3%) | 1 (6.3%) | | |
| High school | 60 | 9 (15.0%) | 19 (31.7%) | 20 (33.3%) | 10 (16.7%) | 2 (3.3%) | | |
| Junior college | 97 | 14 (14.4%) | 34 (35.1%) | 26 (26.8%) | 16 (16.5%) | 7 (7.2%) | | |
| Bachelor degree or above | 9 | 1 (11.1%) | 3 (33.3%) | 3 (33.3%) | 1 (11.1%) | 1 (11.1%) | | |
| Household type | | | | | | | 28.331 | 0.000 |
| Urban residents | 239 | 29 (12.1%) | 76 (31.8%) | 63 (26.4%) | 45 (18.8%) | 26 (10.9%) | | |
| Rural farmers | 108 | 7 (6.5%) | 20 (18.5%) | 25 (23.2%) | 20 (18.5%) | 36 (33.3%) | | |
| Marital status | | | | | | | 32.027 | 0.012 |
| Married | 267 | 31(11.6%) | 74 (27.7%) | 71 (67.7%) | 55 (20.6%) | 36 (13.5%) | | |
| Single | 7 | 3 (42.9%) | 0 (0.0%) | 2 (28.6%) | 1 (14.3%) | 1 (14.3%) | | |
| Divorced | 61 | 1 (1.6%) | 18 (29.5%) | 12 (19.7%) | 8 (13.1%) | 22 (36.1%) | | |
| Widowed | 12 | 1 (8.3%) | 4 (33.3%) | 3 (25.0%) | 1 (8.3%) | 3 (25.0%) | | |
| Occupation before retirement | | | | | | | 51.861 | 0.000 |
| Enterprise employees | 91 | 14(15.4%) | 29 (31.9%) | 24 (26.4%) | 19 (20.9%) | 5 (5.5%) | | |
| Civil servants | 32 | 8(25.0%) | 12 (37.5%) | 4 (12.5%) | 4 (12.5%) | 4 (12.5%) | | |
| Institutions | 92 | 8(8.7%) | 28(30.4%) | 27 (29.3%) | 14 (15.2%) | 15 (16.3%) | | |
| Self-employed | 37 | 3(8.1%) | 11 (29.7%) | 10 (27.0%) | 6 (16.2%) | 7 (18.9%) | | |
| Farmers | 36 | 2(5.6%) | 9(25.0%) | 9 (25.0%) | 8 (22.2%) | 8 (22.2%) | | |
| Migrant workers | 54 | 1(1.9%) | 7(13.0%) | 13 (24.1%) | 12 (22.2%) | 21 (38.9%) | | |
| Others | 5 | 0(0.0%) | 0(0.0%) | 1 (20.0%) | 2 (40.0%) | 2 (40.0%) | | |
| Medical treatment insurance | | | | | | | 64.252 | 0.000 |
| None | 23 | 2 (8.7%) | 3 (13.0%) | 2 (8.7%) | 8 (34.8%) | 8 (34.8%) | | |
| Commercial insurance | 34 | 5 (14.7%) | 16 (47.1%) | 9 (26.5%) | 1 (2.9%) | 3 (8.8%) | | |
| New rural cooperative | 73 | 6 (8.2%) | 12 (16.4%) | 16 (21.9%) | 11 (15.1%) | 28 (38.4%) | | |
| Urban residents medical treatment insurance | 49 | 8 (16.3%) | 10 (20.4%) | 16 (32.7%) | 24 (28.6%) | 1 (2.0%) | | |
| Town worker medical insurance | 148 | 14 (9.5%) | 49 (33.1%) | 37 (25.0%) | 27 (18.2%) | 21 (14.2%) | | |
| Endowment insurance medical insurance | 20 | 1 (5.0%) | 6 (30.0%) | 8 (40.0%) | 4 (20.0%) | 1 (5.0%) | | |

| | | Cognition degree | | | | | | |
|---|---|---|---|---|---|---|---|---|
| Items | Number | Well known | Know | Better known | Little known | Unknown | χ² value | P value |
| Endowment Insurance | | | | | | | 27.857 | 0.020 |
| None | 53 | 3 (5.7%) | 18 (34.0%) | 18 (34.0%) | 5 (9.4%) | 9 (17.0%) | | |
| Agency institution | 117 | 17(14.5%) | 33 (28.2%) | 30 (25.6%) | 23 (19.7%) | 14 (12.0%) | | |
| Urban workers or urban and rural residents | 125 | 14(11.2%) | 29 (23.2%) | 32 (25.6%) | 27 (21.6%) | 23 (18.4%) | | |
| Land expropriated farmers | 26 | 1 (3.8%) | 4 (15.4%) | 5 (19.2%) | 6 (23.1%) | 10 (38.5%) | | |
| Business | 26 | 1 (3.8%) | 12 (46.2%) | 3 (11.5%) | 4 (15.4%) | 6 (23.1%) | | |
| Family monthly income | | | | | | | | |
| <3,000 RMB | 46 | 3 (6.5%) | 6 (13.1%) | 3 (6.5%) | 14 (30.4%) | 20 (43.5%) | 43.312 | 0.000 |
| 3,000–5,999 RMB | 87 | 7 (8.0%) | 25 (28.7%) | 23 (26.4%) | 14 (16.1%) | 18 (20.7%) | | |
| 6,000–9,999 RMB | 110 | 12 (10.9%) | 30 (27.3%) | 37 (33.6%) | 18 (16.4%) | 13 (11.8%) | | |
| ≥10,000 RMB | 104 | 14(13.5%) | 35 (33.7%) | 25 (24.0%) | 19 (18.3%) | 11 (10.6%) | | |

**Table 2 Relationship between family situation and degree of support.**

| | | Degree of support | | | | | | |
|---|---|---|---|---|---|---|---|---|
| Items | Number | Well supported | Support | Better supported | Little supported | Unsupported | χ² value | P value |
| Monthly income | | | | | | | 29.919 | 0.002 |
| <3,000 RMB | 46 | 6 (13.0%) | 27 (58.7%) | 11 (23.9%) | 2 (4.3%) | 0 (0.0%) | | |
| 3,000–5,999 RMB | 87 | 19 (21.8%) | 41 (47.1%) | 26 (29.9%) | 1 (1.1%) | 0 (0.0%) | | |
| 6,000–9,999 RMB | 110 | 36 (32.7%) | 51 (46.4%) | 22 (20.0%) | 1 (0.9%) | 0 (0.0%) | | |
| ≥10,000 RMB | 104 | 38 (36.5%) | 48 (46.2%) | 14 (13.5%) | 2(1.9%) | 2 (1.9%) | | |
| Monthly fee | | | | | | | 57.255 | 0.007 |
| 0–999 RMB | 94 | 20 (21.3%) | 37 (39.4%) | 34 (36.2%) | 3 (3.2%) | 0 (0.0%) | | |
| 1,000–1,999 RMB | 104 | 27 (26.0%) | 65 (62.5%) | 12 (11.5%) | 0 (0.0%) | 0 (0.0%) | | |
| 2,000–2,999 RMB | 61 | 19 (31.1%) | 27 (44.3%) | 13 (21.3%) | 0 (0.0%) | 2 (3.3%) | | |
| 3,000–3,999 RMB | 31 | 4 (12.9%) | 17 (54.8%) | 9 (29.0%) | 1 (3.2%) | 0 (0.0%) | | |
| ≥4,000 RMB | 57 | 29 (50.9%) | 21 (36.8%) | 5 (8.8%) | 2 (3.5%) | 0 (0.0%) | | |
| Degree of child support | | | | | | | 194.993 | 0.000 |
| Well supported | 106 | 62 (58.5%) | 32 (30.2%) | 11 (10.4%) | 1 (0.9%) | 0 (0.0%) | | |
| Support | 167 | 29 (17.4%) | 104 (62.3%) | 30 (18.0%) | 3 (1.8%) | 1 (0.6%) | | |
| Better supported | 62 | 6 (9.7%) | 27 (43.5%) | 28 (45.2%) | 1 (1.6%) | 0 (0.0%) | | |
| Little supported | 4 | 0 (0.0%) | 0 (0.0%) | 3 (75.0%) | 0 (0.0%) | 1 (25.0%) | | |
| Unsupported | 1 | 1 (100.0%) | 0 (0.0%) | 0 (0.0%) | 0 (0.0%) | 0 (0.0%) | | |

health care ($P < 0.05$). From the perspective of family monthly income, the elderly with family income ≥10,000 RMB have more obvious support for the combination of medical and health care than the elderly with family monthly income <3,000 RMB in the control group ($P < 0.05$). In terms of the degree of understanding, the degree of understanding and
**Table 3 Relationship between cognitive level and support level.**

| Items | Number | Degree of support | | | | | $\chi^2$ value | P value |
|---|---|---|---|---|---|---|---|---|
| | | Well supported | Support | Better supported | Little supported | Unsupported | | |
| Know or not | | | | | | | 49.694 | 0.000 |
| Well known | 36 | 22 (61.1%) | 8 (22.2%) | 5 (13.9%) | 0 (0.0%) | 1 (2.8%) | | |
| Know | 96 | 37 (38.5%) | 44 (45.8%) | 13 (13.5%) | 2 (2.1%) | 0 (0.0%) | | |
| Better known | 88 | 21 (23.9%) | 49 (55.7%) | 18 (20.5%) | 0 (0.0%) | 0 (0.0%) | | |
| Little known | 65 | 9 (13.8%) | 32 (49.2%) | 21 (32.3%) | 2 (3.1%) | 1 (1.5%) | | |
| Unknown | 62 | 10 (16.1%) | 34 (54.8%) | 16 (25.8%) | 2 (3.2%) | 0 (0.0%) | | |

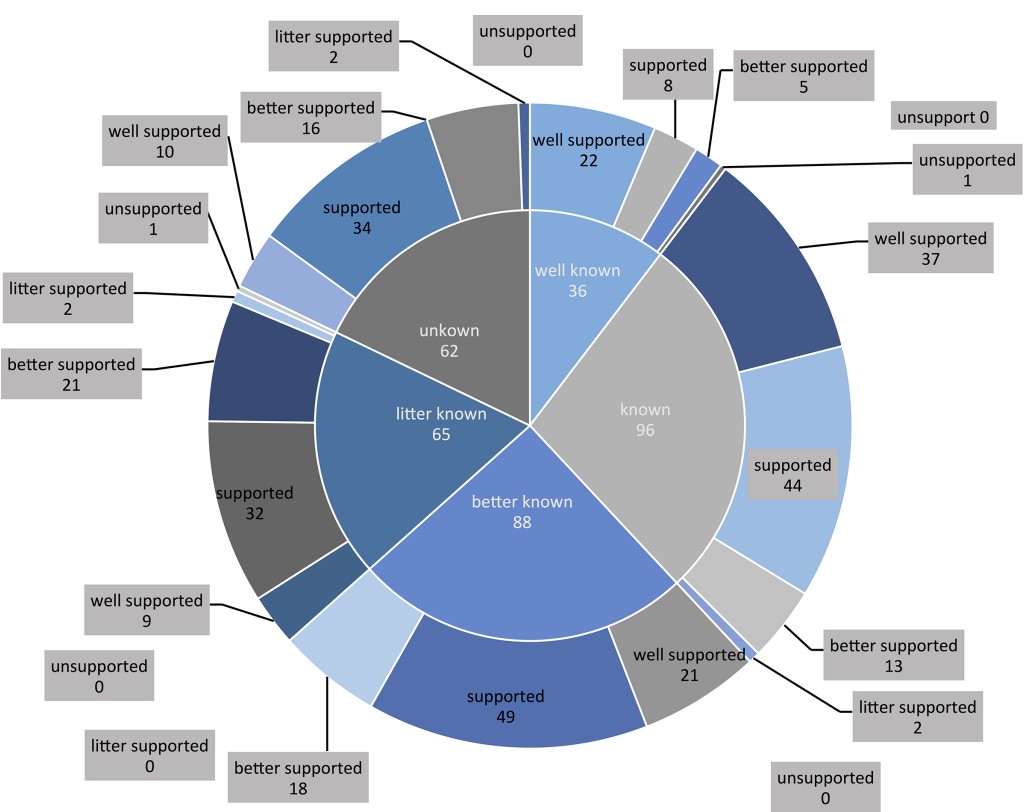

**Figure 2 Relationship between cognitive level and support level.**

support in different degrees are significantly higher than that in the control group ($P < 0.05$) (Table 4).

## DISCUSSION

### The current situation of China's national conditions

At present, China is in the rapid development stage of population aging. The aging problem was serious and causes social anxiety. The elderly have no home and nowhere to go. The disabled and retarded elderly stay in medical institutions and consume medical
**Table 4 Multi-factor Logistic regression analysis of the combination of medical, maintenance and health support.**

| Influence factors | β | $S_{\bar{x}}$ | Wald χ² value | P value | 95% Confidence interval | |
|---|---|---|---|---|---|---|
| | | | | | Upper limit | Lower limit |
| Level of education | | | | | | |
| Primary school the following | 0.000 | | | | | |
| Primary school | −0.053 | 0.389 | 0.018 | 0.892 | −0.815 | 0.709 |
| Junior high school | −0.484 | 0.390 | 1.541 | 0.214 | −1.247 | 0.280 |
| Technical secondary school | 0.200 | 0.564 | 0.126 | 0.722 | −0.906 | 1.307 |
| High school | −0.107 | 0.399 | 0.072 | 0.789 | −0.888 | 0.675 |
| Junior college | −0.352 | 0.394 | 0.797 | 0.372 | −1.124 | 0.420 |
| Bachelor degree or above | −2.576 | 0.873 | 8.697 | 0.003 | −4.287 | −0.864 |
| Household type | | | | | | |
| Urban residents | 0.000 | | | | | |
| Rural farmers | 0.642 | 0.259 | 6.127 | 0.013 | 0.134 | 1.151 |
| Occupation before retirement | | | | | | |
| Enterprise employees | 0.000 | | | | | |
| Civil servants | −0.091 | 0.389 | 0.054 | 0.816 | −0.853 | 0.671 |
| Institutions | 0.023 | 0.279 | 0.007 | 0.935 | -.525 | 0.571 |
| Self-employed | −0.218 | 0.370 | 0.346 | 0.556 | -.943 | 0.507 |
| Farmers | 0.349 | 0.373 | 0.873 | 0.350 | -.383 | 1.081 |
| Migrant workers | 0.914 | 0.331 | 7.646 | 0.006 | .266 | 1.562 |
| Others | 0.363 | 0.865 | 0.176 | 0.675 | −1.333 | 2.059 |
| Family monthly income | | | | | | |
| <3,000 RMB | | | | | | |
| 3,000–5,999 RMB | −0.081 | 0.360 | 0.051 | 0.821 | −0.787 | 0.624 |
| 6,000–9,999 RMB | −0.615 | 0.357 | 2.968 | 0.085 | −1.315 | 0.085 |
| ≥10,000 RMB | −0.778 | 0.364 | 4.563 | 0.033 | −1.492 | −0.064 |
| Know or not | | | | | | |
| Well known | 0.000 | | | | | |
| Know | 0.771 | 0.382 | 4.067 | 0.044 | 0.022 | 1.520 |
| Better known | 1.299 | 0.389 | 11.125 | 0.001 | 0.536 | 2.062 |
| Little known | 2.038 | 0.414 | 24.196 | 0.000 | 1.226 | 2.849 |
| Unknown | 1.747 | 0.415 | 17.764 | 0.000 | 0.935 | 2.560 |

resources. However, more and more elderly people are afraid of getting old because they can't get effective care or the cost of care was too high. Medical and health care will provide full-cycle continuous health management and care for the healthy elderly, high-risk elderly, semi-disabled and disabled elderly in different service scenarios. At present, the state has vigorously advocated the combination of medical and health care at the policy level, and encouraged the provision of one-stop medical and health care services for the elderly (*Tan, 2019*; *Zeng & Liu, 2016*). The outline of "healthy China 2030" plan clearly

proposes to "focus on the whole population and the whole life cycle" and improve the "treatment rehabilitation long-term care service chain".

## Analysis of willingness to combine medical and health care and related factors

Whether the policy was implemented in place, in addition to the strong support of government departments, was related to the willingness of the elderly to combine medical and health care and related factors. In this study, the willingness to combine medical and health care and related factors was studied in Eastern coastal cities in China. It provides a basis for effectively establishing a medical and health care service system in line with Eastern coastal cities in China and establishing a practical, effective and landing integrated medical care service model. From the single factor analysis of Table 1, we can see that the level of education, pre-retirement occupation, registered residence type and family monthly income affect the cognition level of the elderly to medical and healthy care. And this was consistent with the conclusions of Wu Li group (*Jiang, Dong & Li, 2020*; *Wu et al., 2020*). The reason for the analysis was that for this group with high education level, their pre-retirement career will be relatively stable and their family monthly income will be higher. Such groups will also pay more attention to which medical care method was more suitable for themselves. This was also confirmed by the family situation and degree of support in Table 2: the family monthly income was positively correlated with the degree of support for the combination mode of medical and health care, which was also confirmed by the research of *Liu, Shen & Chen (2018)*. It showed that the elderly's support for medical and health care services was restricted by their economic ability. Their family income was high, and they have a higher ability to bear and pay for medical care and health.

The income level was usually positively correlated with the education level. The elderly with higher education level will timely evaluate their health needs and respond positively (*Zhuang, Jiang & Zhang, 2016*). The study found that the elderly with registered residence as urban residents were more familiar with the medical maintenance mode while the suburban residents knew less. The reason was that the medical care market in the urban area has developed rapidly and the information has been obtained more thoroughly.

On the other hand, affected by the baby boom in the 1960s and the family planning policy in the 1980s, the family aging structure has developed abruptly. In the face of the care work of four elderly people, urban residents with limited accompanying care energy more choose to support the medical care model, and the elderly were also affected by the surrounding population. Due to the pressure of public opinion, more children still chose family pension. Therefore, suburban farmers have limited sources of information on the combination of medical and health care. The degree of children's support for the medical care model affected the elderly's cognition and support for the medical care model, which was consistent with the research conclusions of *Liu et al. (2019)*.

Through multivariate logistic regression analysis, from the perspective of educational level, compared with the control group with education below primary school, the elderly with bachelor degree or above are more willing to support the combination of medical care and health care. The registered residence type was more obvious than that of the urban

residents. In terms of occupation, compared with the enterprise employees in the control group, the elderly who were employed as migrant workers before retirement were more willing to support the combination of medical care and health. From the perspective of family monthly income, the elderly with family income ≥10,000 RMB have more obvious support for the combination of medical care and health than the elderly with family monthly income <3,000 RMB in the control group. In terms of the degree of understanding, the degree of understanding and support in different degrees are significantly higher than that in the control group.

## Highlights and shortcomings

In this study, with the help of a typical community health service center, one of the first batch of 50 "national pilot units for the combination of medical care and maintenance" as a carrier, we have an in-depth understanding of the development of combine medical and health care. However, in this study, although 347 valid samples were successfully obtained, which is representative in the eastern coastal cities of China, the sample size still needs to be expanded if it is extended to the whole country. In the future research, information technology can be effectively used to expand the sample size through non-contact methods such as emails and text messages. In addition, there is a bias in the survey of the elderly with dialects. Although the staff will follow the explanation during the survey, the staff will have a certain inductiveness to the elderly, and there may be a certain bias in the results.

## Suggestions

Increase investment in elderly activity centers, and actively carry out elderly activities that were beneficial to physical and mental health. Learning health care knowledge, providing healthy diet and spiritual comfort to alleviate the pressure of family and social elderly care, this was an important way to achieve healthy elderly care and healthy aging. It was also necessary to strengthen the training of professional talents and improve the level of elderly care in communities or institutions. The government and elderly care institutions should do a good job in the planning and construction of institutional elderly care in accordance with the demographic characteristics of the local elderly and the development level of elderly care services. And build a group of new elderly care institutions and fully functional elderly care institutions with medical care, elderly care, nursing care, rehabilitation and end-of-life escort. And strengthen the training of professionals with a combination of medical care and nursing, improve the level of institutional care for the elderly, provide convenient and high-quality medical, rescue, rehabilitation, care, comfort and other services for the elderly, sickly, and disabled elderly, and provide satisfactory material and spiritual services.

## CONCLUSION

In this study, the willingness and related factors of the combine medical and health care were studied in the eastern coastal cities of China, in order to effectively establish a localized combine medical and health care in line with the social conditions, strategies and

institutional background of the eastern coastal cities in China. It provides a basis for meeting the dual needs of the elderly for medical care and old-age care.

Based on the results of this study, we believe that it is necessary to increase investment in elderly activity centers, actively carry out activities that are beneficial to the physical and mental health of the elderly, and improve the level of elderly care in communities and institutions. The government and elderly care institutions should establish fully functional elderly care institutions such as medical care, elderly care, nursing, rehabilitation, and hospice care. To sum up, it is necessary to improve the service level of elderly care institutions, provide the elderly with convenient and high-quality medical, rescue, rehabilitation, nursing, comfort and other services, and provide satisfactory material and spiritual services.

## ACKNOWLEDGEMENTS

The authors of this study thank the Zhejiang Hospital Ethics Committee for their support, two communities and eight nursing homes for their support, and all surveyed elderly people and staff for their support.

### Funding

This work was supported by the Health Science and Technology Plan of Zhejiang Province in 2021 (No. 2021KY008) and the General Project of 2021 Provincial Soft Science Research Program (No. 2021C35095). The funders had no role in study design, data collection and analysis, decision to publish, or preparation of the manuscript.

### Grant Disclosures

The following grant information was disclosed by the authors:
Health Science and Technology Plan of Zhejiang Province: 2021KY008.
General Project of 2021 Provincial Soft Science Research Program: 2021C35095.

### Competing Interests

The authors declare that they have no competing interests.

### Author Contributions

- Yehong Wei conceived and designed the experiments, performed the experiments, prepared figures and/or tables, authored or reviewed drafts of the article, and approved the final draft.
- Yanxiang Sun conceived and designed the experiments, performed the experiments, prepared figures and/or tables, authored or reviewed drafts of the article, and approved the final draft.
- Yuting Li performed the experiments, analyzed the data, prepared figures and/or tables, authored or reviewed drafts of the article, and approved the final draft.
- Xufeng Chen performed the experiments, analyzed the data, authored or reviewed drafts of the article, and approved the final draft.

- Yingyu Wu performed the experiments, authored or reviewed drafts of the article, and approved the final draft.
- Xindi Fang performed the experiments, authored or reviewed drafts of the article, and approved the final draft.
- Ruichen Zhu conceived and designed the experiments, authored or reviewed drafts of the article, and approved the final draft.

## Human Ethics

The following information was supplied relating to ethical approvals (*i.e.*, approving body and any reference numbers):

Zhejiang Hospital granted ethical approval to conduct research in its facilities (Ethical application Reference: 20121 Preliminary Review No. (75K)).

## Data Availability

The raw measurements are provided in the Supplemental Files. These data mainly carry out multi-factor and single-factor analysis on old-age care factors, aiming to establish an integrated service innovation mode of medical treatment, maintenance and health care suitable for the social conditions, strategies and institutional background of the region by analyzing the willingness of the elderly in the region to combine medical treatment, nursing, health care and related factors.

## Supplemental Information

Supplemental information for this article can be found online at http://dx.doi.org/10.7717/peerj.14004#supplemental-information.

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
