# Peer review of "Investigation and research on elderly people’s willingness to combine medical and health care and related factors in coastal cities in eastern China"

_PeerJ, doi:10.7717/peerj.14004_

## Round 0.1 · original submission · Major Revisions

Thank you for submitting the manuscript to PeerJ. It has been reviewed by experts in the field and we request that you make major revisions before it is processed further.

We look forward to hearing from you soon.

Best wishes,

Badicu Georgian, Ph.D

·

Basic reporting

Authors reported on an interesting survey on elderly people’s willingness to combine medical and health care and related factors. The manuscript is generally well-written, but the introduction, rationale, objectives/hypotheses and methodology are not clearly presented. Also, lack of illustration is pretty obvious and could bore the readers. Given the importance of the subject for general health, the manuscript should give more info on the matter. Although the study results are promising and important for the field of health promotion, I have some major methodological concerns and other issues that the authors need to address before I can accept the manuscript for publication.

Specific comments

Abstract
• Authors should consider adding more info on the methods, and measurement units (i.e., questions used in the questionnaire) they used in the study. Was the questionnaire verified and validated previously or a version of a standardised one?

• The Results section does not mention anything about the male/female composition of the group.
• The info should be better presented as to raise the interest of the readers.
• The key words must be different from the title words.
• The title should refer to this country/region as the whole article is based on Chinese studies.

Introduction

• The information given by the authors is not sufficient to create the background for this important matter to health. There are only 3 references in this section and they are not enough. Can the authors maybe provide some more background on the importance of informing people early in life about this serious problem they will confront later?

• The rationale for examining this problem should be mentioned more clearly in this section. Why did the authors choose to examine it? Why did the authors choose to examine some factors and not other factors such as the quality of life, personal beliefs or accessibility of medical and health care? At least the reader should be given a background on how each of the identified factors are important and informed on the novelty of this study.

• The article innovation should be presented in the Introduction. Describe what the research gap of the paper is and what is new. Please describe the links between the research gap and the goal of the article.

• Lines 37-38 are not enough to make your study of sound scientifically importance. Please provide a subsection here with the title RESEARCH QUESTIONS/HYPOTHESES which would be more convincing about the importance of the study.

Experimental design

Participants and methods

• The number of the questions used in the questionnaire or some examples of them are nowhere to be found, as a reader can assume from the title that there are the questions somewhere to be presented.

• It is also necessary to describe and incorporate empirical evidence of validity and reliability of the questionnaire which is applied in this research. This information is essential to assume the soundness of the obtained results.

• There are doubts for readers about the choosing of the proper questions. The procedure is not explained. What were the questions? A reader can only guess from the Results section what were the questions. All these need to be spelled out clearly in the methods section.

• It is not clear how the authors applied the study design in terms of control group. Could this be clarified?

Validity of the findings

The Results section should be reorganised as to follow each hypothesis/research question or objective. Authors need to write key findings focusing on each one of these after being stated.

An illustration using different colours for numbers would be helpful in presenting the results as to raise the interest of the readers. The plain explanations here are somehow boring and repeated form the tables.

In tables 1, 2, 3 the items are puzzled, seem to be missing and difficult to follow.

The Discussion section is weak as there are few references here, too.

It would be better to have seen more use of terms like 'originality' and 'significance'. Identify what is new in this study that may benefit readers or how it may advance existing knowledge or create new knowledge on this subject. There should be a clear conclusion on why the research findings are significant for health subject and could be used for the help of people in this situation.

It should be presented a model from table 1, not just mention the possibility of creating one.

The study could be extended by adding the number of people in need of such care, the number of existing care institutions and the number of the institutions-to-be. It could be interesting if the authors presented the number of total population in this region, the number of elderly in need for care.

Articles usually do not use the word "we" and regularly use passive verbs. Long sentences are to be avoided.

Research limitations and existing problems are not presented.

Reviewer 2 ·

Basic reporting

Need recheck for English from a fluent English speaker
Add detail in introduction section regarding topic, previous literature, improve rationale of study

Experimental design

Research question defined but need improvement
Rigorous investigation performed
Add separate heading for ethical statement
Add sample size calculation
Add pilot scale study

Validity of the findings

Create 1 or 2 figures related to study results
Add Limitation and strength of study in a separate heading
Add conclusion
Check journal guidelines and format manuscript accordingly

---

## Round 0.2 · Minor Revisions

In the Introduction, I think it is necessary to present other studies similar or close to this topic from different countries, not only from China.

·

Basic reporting

Thank you for providing this comprehensive work.
The authors have presented an improved version of the manuscript.

The introduction provides a proper background of the topic. The sections/titles have been improved.
The quality of the images is good enough, but I don’t know if the reviewing version has lower resolution than the final version. If not, images should have better resolution in its final size.
It seems that the English is technically correct.

Experimental design

The experimental design meets the scope of the journal, and it is relevant to the community.
Methods are described detailed enough.

Validity of the findings

The results and the conclusions are quite interesting and well-discussed. All data are provided.

The authors have adequately addressed all my comments. I have no further suggestions.

Reviewer 2 ·

Basic reporting

Accept in current form no further comments

Experimental design

Accept in current form no further comments

Validity of the findings

Accept in current form no further comments

Additional comments

Accept in current form no further comments

---

## Round 0.3 · accepted · Accept

Currently, the article is acceptable for publication. Congratulations!

We look forward to hearing from you soon.

Best wishes,

Badicu Georgian, Ph.D